# Structure-Property Relationships of Polyamide 12 Grades Exposed to Rapid Crack Extension

**DOI:** 10.3390/ma14195899

**Published:** 2021-10-08

**Authors:** Mario Messiha, Andreas Frank, Jan Heimink, Florian Arbeiter, Gerald Pinter

**Affiliations:** 1PCCL GmbH, 8700 Leoben, Austria; andreas.frank@pccl.at; 2Evonik Operations GmbH, 45772 Marl, Germany; jan.heimink@evonik.de; 3Department Polymer Engineering and Science, Montanuniversitaet, 8700 Leoben, Austria; florian.arbeiter@unileoben.ac.at (F.A.); gerald.pinter@unileoben.ac.at (G.P.)

**Keywords:** small-scale steady state (S4) test, rapid crack propagation (RCP), polyamide 12, structure-property relationships, strain-rate effects

## Abstract

Thermoplastic materials have established a reputation for long-term reliability in low-pressure gas and water distribution pipe systems. However, occasional *Slow Crack Growth* (SCG) and *Rapid Crack Propagation* (RCP) failures still occur. SCG may initiate only a small leak, but it has the potential to trigger RCP, which is much rarer but more catastrophic and destructive. RCP can create a long, straight or meandering axial crack path at speeds of up to hundreds of meters per second. It is driven by internal (residual) and external (pressure) loads and resisted by molecular and morphological characteristics of the polymer. The safe installation and operation of a pipe throughout its service lifetime therefore requires knowledge of its resistance to RCP, particularly when using new materials. In this context, the RCP resistance of five different polyamide (PA) 12 grades was investigated using the ISO 13477 *Small-Scale Steady State* (S4) test. Since these grades differed not only in molecular weight but also in their use of additives (impact modifiers and pigments), structure-property relationships could be deduced from S4 test results. A new method is proposed for correlating these results more efficiently to evaluate each grade using the crack arrest lengths from individual S4 test specimens.

## 1. Introduction

To determine the resistance against *Rapid Crack Propagation* (RCP) of newly developed plastic pipe grades, the *Full-Scale* (FS) test, standardized in ISO 13478, has become the “gold standard” for product qualification. It measures a critical pressure value (*p*_c,FS_) above which RCP can occur under operating conditions for pressurized pipes in service. However, it requires 25 m long pipe specimens of up to 500 mm diameter. Since several FS tests are needed to evaluate the RCP behavior of a given material, this method is expensive and time-consuming.

These disadvantages were mitigated by the development of an accelerated laboratory test, the so-called *Small-Scale Steady State* (S4) test standardized for thermoplastic pipes in ISO 13477 [1]. The S4 test initiates a rapid crack in a pipe sample only 7 to 8 pipe diameters (*D*) long. As in the FS test, pipes are tested at a specific temperature (e.g., 0 °C) and pressurized with air. RCP is initiated by the impact of a chisel-ended striker close to one end of the pipe. To prevent excess flaring of the pipe walls while allowing the compressed air to escape freely, a rigid cage surrounds the sample, while internal baffles retard axial decompression and ensure a relatively steady pressure in each compartment between them [2].

The result of each S4 test is classified as *propagation* if the crack extends to a length *a* within the gauge section of at least 4.7*D*, or *arrest* if 0.7*D* < *a* < 4.7*D*. Cracks shorter than 70% of a pipe’s diameter are regarded as being too highly influenced by gauge section end effects, e.g., insufficient internal gas volume. A series of S4 tests can determine either a *critical pressure* (*p*_c,S4_) for pipe at a specified temperature or a *critical temperature* at a specified pressure (Figure 1). In either case, however, *false arrest* points can appear within the propagation regime (see Figure 1) resulting in a bell-shaped (*cloche*) crack-length curve and complicating the evaluation of critical pressure or temperature [2].

The present work developed a more efficient procedure for characterizing the RCP behavior of morphologically different PA12 grades. Various S4 tests were carried out to characterize critical RCP/crack arrest transition temperatures and pressures, and to more securely establish structure-property relationships. To better understand underlying physical processes, detailed post-mortem examination of fracture surfaces was carried out.

## 2. Experimental

The five PA12 grades selected for this study (Figure 2) were part of a systematic series of unplasticized grades that differed primarily in their molecular weight (*M*_W_) and/or the use of specific additives, such as an impact modifier (IM) or an inorganic pigment (PGM). Four of the five grades, coded with the letter “C”, are highly viscous compounds based on PA12-0 (Figure 2). C3-nc is a natural colored (nc) compound without impact modification. Adding an IM to it yields C4-im-nc, while additional coloration results in C2-im-pgm. Finally, C5-im-pgm can be viewed as a derivative of C2-im-pgm, modified to increase molecular weight.

Weight average molecular mass *M*_w_ was measured using Size Exclusion Chromatography (SEC). Samples taken from extruded pipes of the given materials were dissolved in hexafluoroisopropanol (HFIP) and potassium–trifluoroacetate at room temperature. Table 1 summarizes relevant properties of the selected materials; for clarity, *M*_w_ values are normalized against that of PA12-0.

Each grade was first characterized for RCP resistance using the S4 test according ISO 13477 [1]. Extruded pipe specimens with a length of 835 mm, an outer diameter of *D* = 110 mm and a Standard Dimension Ratio (SDR) 11 were used. Applied pressures ranged from 1.5 bar to 8 bar at temperatures between 0 °C and 45 °C and the impact speed was 16 m/s. Furthermore, a newly modified data evaluation process, which additionally accounts for the crack length *a* in relation to the to the maximum possible crack length (*a*_max_) of 758 mm (i.e., from the center of the impact position to the end of the gauge length), is suggested. Fracture surfaces were studied using a Tescan Vega II Scanning Electron Microscope (Tescan Brno, Brno, Czech Republic).

## 3. Results and Discussion

Critical pressure results for the five PA12 grades are shown in Figure 3. Solid symbols represent tests with valid RCP initiation according to ISO 13477 [1], while hollow symbols represent valid RCP arrests. At *T* = 0 °C, the grades can be grouped as follows:Those with low critical pressures of ~3.2 bar: these include neat PA12-0 as well as C2-im-pgm;Those with slightly higher *p*_c,0°C_ values of 4.1–4.4 bar: C4-im-nc and C5-im-pgm; andC3-nc, having the remarkably high *p*_c,0°C_ of ~8 bar.

At temperatures above 20 °C the critical RCP pressure increases significantly for all materials tested. Only PA12-0 still exhibited valid RCP at *T* > 25 °C and pressures of up to 12 bar, with no crack arrest seen even at 45 °C—which exceeds the *T*_g_ (40 °C). This indicates a very high critical S4 temperature *T*_c,S4_ and very poor RCP resistance. Compounded grades, on the other hand, exhibit *T*_c,S4_ values of around ambient temperature.

**Figure 3 materials-14-05899-f003:**
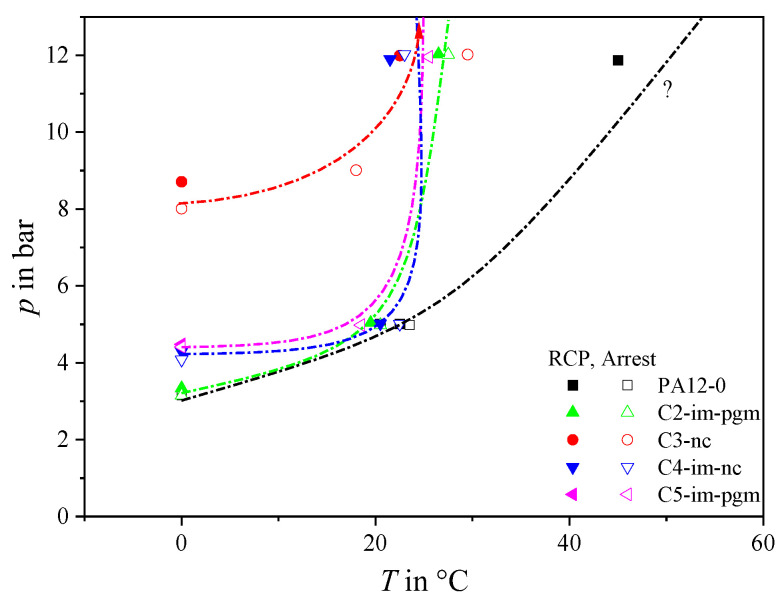
RCP characteristics of various PA12 grades measured via S4 tests.

The chain lines shown for each material in Figure 3 sketch boundaries between RCP arrest and propagation regions. However, there are not enough data points to identify for every material, with adequate precision, either an ISO 13477 [1] critical pressure at a given temperature (e.g., C3-nc at *T* = 0 °C) or a critical temperature at a given pressure (e.g., PA12-0 and C5-im-pgm at *p* = 12 bar). To do so, as Figure 2 illustrates schematically for just one material, a significant number of additional tests would be needed.

To maximize the data yield from each test the measured arrest length of each crack was processed numerically, as well as being classified as arrest/propagation (see Figure 4 and Figure 5). Plotting at least three crack lengths as a function of test pressure at a given temperature (e.g., at 0 °C, Figure 4) allows the critical pressure to be extrapolated by linear regression. For example, instead of starting at extreme pressures and successively decreasing the applied *p* at a constant *T* to obtain the point of transition from RCP to crack arrest, it is proposed to test first at very low *p* and then at very high *p*. The critical S4 pressure point can then be estimated by intra- or extrapolation and refined successively by additional intermediate tests. If possible, it is recommended to base the linear regression primarily on data points between 0.7*D* < *a* < 4.7*D*, in order to avoid effects from initiation as well as reflection of decompression waves and end-cap constraints.

Using this method yielded slightly different *p*_c,0°C_, *T*_c,5bar_, and *T*_c,12bar_ values (Figure 5 and Figure 6) from those using the ISO standard method. Each grade except PA12-0 could be characterized at 0 °C, 5 bar, and 12 bar (Figure 5b). The use of crack length data also makes it easier to exclude false arrest points that might significantly affect the critical pressure result (e.g., C2-im-pgm in Figure 4).

Table 2 compares transition values obtained using this method with those from the ISO standard procedure. Clearly the same amount of S4 test data can now yield more *T*_c,5bar_ and *T*_c,12bar_ values.

Transition temperatures extrapolated using crack length measurements are plotted in Figure 6. Except for PA12-0—having only two transition point data—transition lines were determined via exponential fitting, allowing standard S4 failure lines to be extended beyond the test data envelope (e.g., *T*_c,5bar_ of C3-nc). The two evaluation methods yield very similar results (cf. Figure 3 and Figure 6) showing that PA12-0 and C2-im-pgm fall short of the remaining compounds, becoming susceptible to RCP at lower applied pressures. While C2-im-pgm becomes resistant to RCP above ambient temperature, PA12-0 remains susceptible over a wider temperature range. In contrast, C3-nc exhibits an extraordinary RCP resistance. At *T* = 0 °C, RCP only occurs at much higher pressures than in the other grades. This implies that significantly increasing *M*_W_ by compounding (e.g., from PA12-0 to C3-nc, Figure 2) benefits RCP resistance. Krishnaswamy et al. [4] and Argyrakis [3] reported similar observations for PE-HD: high molecular weight and narrow molecular weight distributions appear to be important to superior RCP resistances in PE-HD.

In contrast, comparing C3-nc and C4-im-nc indicates that impact modifier has a negative effect on RCP resistance. Rubbers are widely used to toughen polymers in critical applications that might expose them to impact loads. In theory, the soft particles absorb much of the input impact energy, either by stretching the rubbery material, or by promoting multiple crazing, shear yielding or the combination of both [5]. The contribution of each mechanism to toughen the rubber-matrix system depends on a number of variables, such as the rubber particle size, distribution and concentration [6]. However:(1)Impact modifiers generally increase the *initiation resistance G*_Ic_(*t*) to rapid loading of a crack but do not influence resistance to its propagation [7,8,9,10];(2)Impact modifiers are themselves time-dependent elastomers undergoing significant strain-rate hardening [5,10,11].

It can, therefore, be assumed that rubber particles have enough time during static and quasi-static load conditions (i.e., low and moderate local strain rates) to plastically deform and cavitate, enhancing the resistance to (quasi-) static crack initiation *G*_Ic_ through extensive shear yielding or multiple crazing phenomena. If the applied (quasi-) static crack driving force *G* reaches the critical *G*_Ic_ a crack starts to extend at low crack speeds $\dot{a}$. Thereby, low crack opening strain rates prevail at areas surrounding the crack tip, so that incorporated elastomers get easily strained and continue to absorb large amounts of energy even during slow crack extension [5]. In contrast, a structure exposed to highly dynamic, time-dependent load conditions *G*(*t*) experiences local deformations at extremely high rates. Once *G*(*t*) becomes equal to or bigger than the time-dependent initiation fracture toughness *G*_Ic_(*t*) of a rate-sensitive material, a rapid crack is initiated. In that context, time *t* is substitutable with the applied impact speed *v* during an abrupt loading. Considering Figure 7, it becomes clear why an improvement of fracture initiation properties due to rubber toughening of rate-sensitive plastics is ineffective at very high impact speeds (e.g., S4 tests). As *G*_Ic_(*v*) is possibly reduced to a minimum value [12], the fundamental rapid fracture resistance of different pipe grades during S4 testing at 16 m/s load speed can be considered to be primarily dependent on the dynamic crack propagation resistance *G*_Id_($\dot{a}$) after crack initiation.

Subsequently, the soft rubber parts of the microstructure at the vicinity of the fast-running crack tip have no time to follow the governing local crack opening deformations. Even worse, bearing the time-temperature principle in mind, it may be well suggested that elastomeric parts could pass through the glass transition when a critical local strain rate is exceeded, which is equivalent to the rubber’s *T*_g_ or tough-brittle transition temperature (*T*_TB_). In that case, the hardened rubber particles might act as additional stress concentrators, at which rapid cracks initiate and propagate more easily. As a result, the incorporation of tougheners may feature a significant reduction of the RCP resistance during S4 testing [10]. This hypothesis is validated by the S4 results of C3-nc and C4-im-nc, where the systematic addition of an IM leads to remarkably lower critical *p*_c,S4_ values (Figure 6).

On the same basis, the pigmentation of materials exposed to crack driving forces can correspond to the addition of inherent defects and localities of stress concentrations at which cracks are preferentially launched. Several studies confirm that the improvement of esthetical properties of different materials by coloration is often accompanied by significant reductions of their fracture toughness [14,15,16,17]. In that manner, the addition of an inorganic pigment to C4-im-nc resulting in C2-im-pgm, yields to a further decrease of *p*_c,S4_ from approximately 4.2 to 3.2 bar (or even less, if false arrests are accounted for). Simultaneously, the critical *T*_c,S4_ is slightly shifted towards higher values—from 23 °C to 27.5 °C—which further affirms the rather negative impact of coloration on rapid fracture behavior. Finally, C5-im-pgm, which equals a high-molecular weight C2-im-pgm, achieves improved *p*_c,S4_ values of 4.5 bar and slightly lower *T*_c,S4_ values of 25 °C. This corresponds to the counterbalancing effect of an enhanced *M*_w_ on a material’s RCP resistance.

Studying the overall shape of the S4 failure regimes presented in Figure 6 it can also be seen that both grades, which do not contain a rubber part, exhibit a successive increase of RCP resistance with rising temperatures. In contrast, the three impact modified grades hardly show any kind of improvement in terms of RCP resistance with increasing temperatures up to 0 °C. However, once a critical temperature around 10–20 °C is reached, the impact modifier seems to go through a brittle-tough transition at underlying strain rates and quickly transfers from an inherent defect to an energy absorber, explaining the rather sharp transition observed in S4 results of C2-im-pgm, C4-im-nc, and C5-im-pgm.

Fracture surfaces of S4 tested PA12 samples are presented in Table 3. Macroscopic overview images of all grades (Table 3a) reveal a major difference between the general fracture appearance of non-toughened and toughened grades. Basically, non-toughened grades PA12-0 and C3-nc exhibit a very rough surface, with structural discrepancies. Contrarily, PA12 grades with integrated soft particles present smoother fracture surfaces. Furthermore, a distinction between two noticeable microscopic structures can be made for all grades: regions of distinctive *river lines* that look like deep carvings inside the fractured surfaces (Table 3b), as well as a dominating flat failure region (often denoted as mirror zone [18]), which appears featureless at low magnification images (Table 3c). At this mirror zone intense crack acceleration might be expected, regardless of the imposed load rate. Although river lines are commonly observed on RCP fracture surfaces [3,19], no universal theory could be found, describing how this river pattern of radiating lines is caused. Basically, river lines are believed to originate from the interconnection of micro-cracks at different fracture surface levels of cleavage plains with different orientations with the main fracture plane [20]. Hayes et al. [21] observed river markings to be more pronounced if polymers experience a mixed mode loading, in particular Mode I and Mode III. As the main crack moves across the component, micro-cracks coming from different cleavage planes have to rotate to keep the continuity of RCP. Thus, river markings are dependent on the micro-structure of the material and become significantly prominent with increasing Mode III [20,21]. From Table 3b it can be seen, that river lines also vary on a microstructural scale. Particularly, C4-im-nc and C5-im-pgm display a spherulite-like marking of different sizes around river lines, which may be provoked by either the incorporated IM or pigment. Surprisingly, the structure of C2-im-pgm is quite similar to that of C3-nc, which does not contain IM or pigments. A magnified view on the featureless fracture region (compare Table 3a–c) reveals a reversed tendency with regard to aforementioned macroscopic fracture appearances. In that context, PA12 grades that contain an impact modifier highlight a very coarse surface with many irregularities and bumps, especially in combination with color pigments (i.e., C2-im-pgm and C5-im-pgm), whereas PA12-0 and C3-nc are relatively smooth in comparison.

In order to analyze the aforementioned river lines more deeply, the scanning electron beam used during SEM measurements was angled about 45°–52° to fracture surface. In doing so, it became possible to take a closer look underneath river markings on RCP fracture surfaces. An evident fibrillation was found beneath each observed river line in almost all grades—an example of C2-im-pgm is given in Figure 8a. Principally, there are two major views, advocated in the scientific community, with regard to the actual failure mechanism during rapid fracture events. The first perspective considers chain scission of covalent backbone bonds as dominant failure mechanism as reported by Donald and Kramer [22,23], Plummer and Kausch [24], as well as Deblieck [25]. This position is based on the fact that the crack speed is too high to promote chain reptation and subsequent disentanglement. The second viewpoint, however, considers chain disentanglement to be still the dominant mechanism, as it is the case for slow crack growth (SCG) [22,23,26,27,28,29]. In contrast to SCG, however, which is guided by mechanical creep and chain slipping against secondary forces (e.g., van der Waal, hydrogen bonds, etc.) of neighboring chains, chain disentanglement is assumed to be promoted by additional adiabatic heating processes [12,30,31,32] during fast fractures. The idea of a thermal decohesion mechanism during RCP assumes that adiabatic conditions are certainly obtained at high crack speeds due to the low thermal conductivity of polymers. This was reported by Leevers [12] for PE-HD, but also for brittle bulk metallic glass that exhibited a maximum crack velocity of ~800 m/s [33]. Considering Figure 8a, such fibrillated structures seem to be highly improbable, if RCP is solely governed by chains scission. Yet, an adiabatic decohesion mechanism [12,30,32], would predict thermo-mechanically activated fibrillation and cavitation. In agreement with aforementioned fundamental observations of river markings, a conceivable explanation for river lines in PA12 could be that they were regions of rapidly extending crack fronts, which eventually stopped at an instance of time, before jumping into a neighboring crack plane (Figure 8b). Reason therefore could be the superposition of different reflected stress waves at the crack tip, varying stress triaxiality levels along the thickness of the pipe, as well as inhomogenously distributed weak points (e.g., crystal defects, contaminations, pigments or hardened impact modifiers). Only if adiabatic decohesion is governing RCP on a molecular level, the abrupt change of the main crack plane would leave behind a ridge of fibrillar structures, that is not able to continue participating in the physical crack extension by thermal decohesion [12,30,32]. With a new hot craze-crack front passing by in a “new” main crack plane, fibrils beneath river lines solidify, freezing in the observed structures in Figure 8a.

Another feature indicating the presence of high temperature localities during RCP might be seen in uniform-width, almost-straight lines fibrillar structures that could be observed on the high-magnification images of most PA12 grades (Figure 9). The length of this thin fibrils deviates from nano-meter to micro-meter scale, whereas the thickness remains essentially constant (about a few hundred nano-meters). At the current stage it is not clear how these structures were built and why they did not collapse back into the melt layer, if an adiabatic decohesion mechanism is true for RCP fractured PA12 grades (e.g., as thermoplastic fibers would do in the hot air stream from a hair-dryer). However, it is clear, that huge amounts of thermal energy are necessary to drain these very thin structures from a craze zone or the bulk during fracture. Additionally, perhaps if they were thicker they would contract back to the melt zone, because they would retain heat for long enough to do so, yet, due to their very thin nature solidification may occur very fast [34].

## 4. Conclusions and Outlook

This study has highlighted relevant structure-property relationships regarding the resistance against *Rapid Crack Propagation* (RCP) in morphologically different polyamide (PA) 12 grades. The RCP performance was measured using the *Small-Scale Steady State* (S4) test according to ISO 13477. An obvious improvement of the RCP resistance was observed with increasing molecular weight *M*_W_, while the incorporation of inorganic pigments reduced the rapid fracture toughness of PA12. Furthermore, rubber-toughening of PA12 grades was also found to decrease the RCP resistance, due to underlying strain-rate effects. In contrast to the general believe that chain scission mechanism is governing physical processes on a molecular scale during RCP, fractographic analysis of S4 fractured pipe samples indicate the existence of high temperature, probably due to adiabatic conditions. In this regard the present work may be viewed as the first step to explain the complex relationship between rapid fracture performances of PA12 grades and underlying fracture mechanisms, such as thermo-mechanical chain disentanglement by means of adiabatic decohesion. Another key element of this research was the proposal of a novel evaluation concept of the S4 test results to improve the determination of critical temperature and pressure values on the basis of a significantly reduced amount of S4 tests.

## Figures and Tables

**Figure 1 materials-14-05899-f001:**
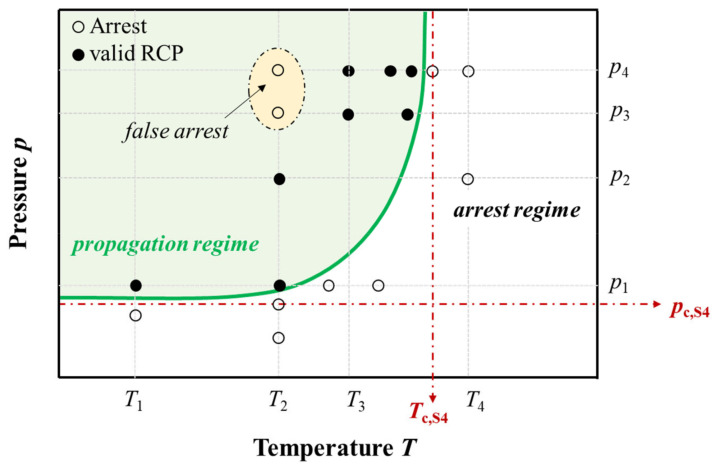
Schematic of crack propagation and arrest regimes for RCP during S4 testing as well as false arrest points within the propagation regime (according to [3]).

**Figure 2 materials-14-05899-f002:**
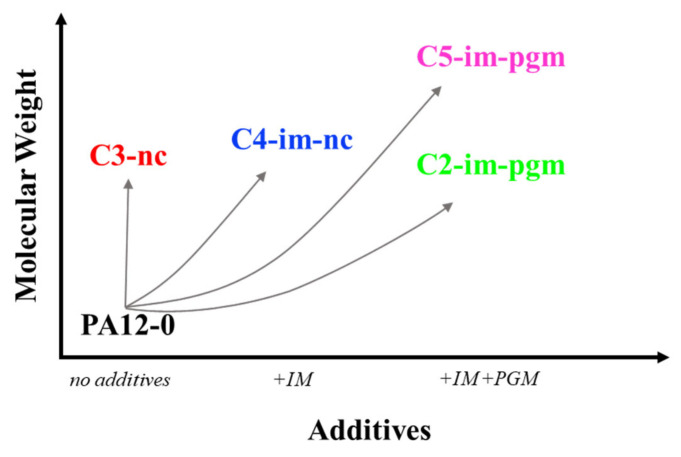
Systematically developed series of unplasticized PA12 grades.

**Figure 4 materials-14-05899-f004:**
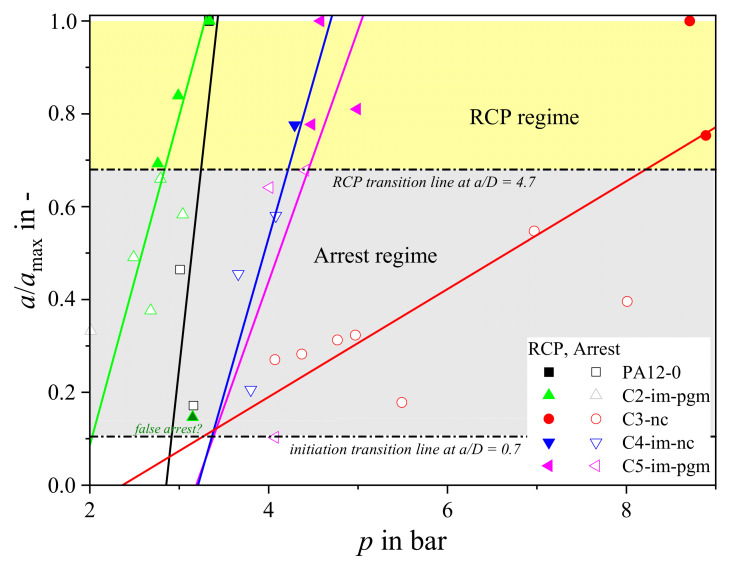
Modified evaluation of the critical pressure values of all grades at 0 °C.

**Figure 5 materials-14-05899-f005:**
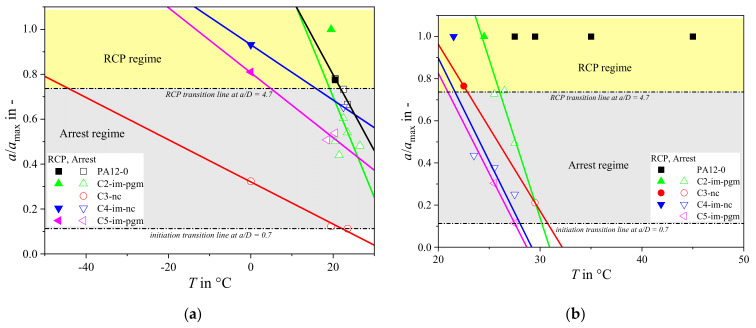
Modified evaluation of the critical temperature values of all grades at (**a**) 5 bar and (**b**) 12 bar.

**Figure 6 materials-14-05899-f006:**
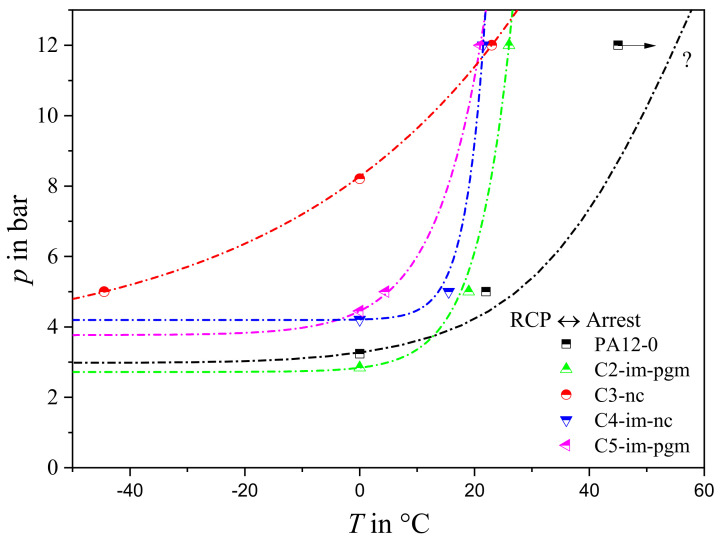
Rapid fracture behavior of PA12 grades as acquired from S4 tests according to the modified evaluation with predicted RCP/crack arrest transition points.

**Figure 7 materials-14-05899-f007:**
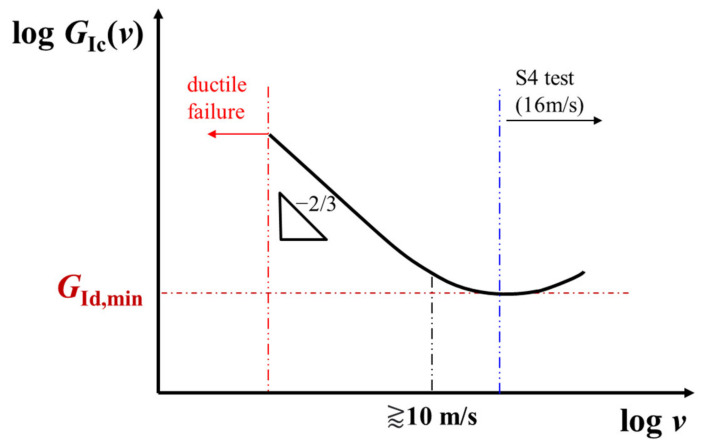
Schematic illustration of a time-dependent initiation energy release rate *G*_Ic_(*v*) as a function of the load rate *v* for rate-sensitive materials with dominating embrittlement (in accordance to [13]).

**Figure 8 materials-14-05899-f008:**
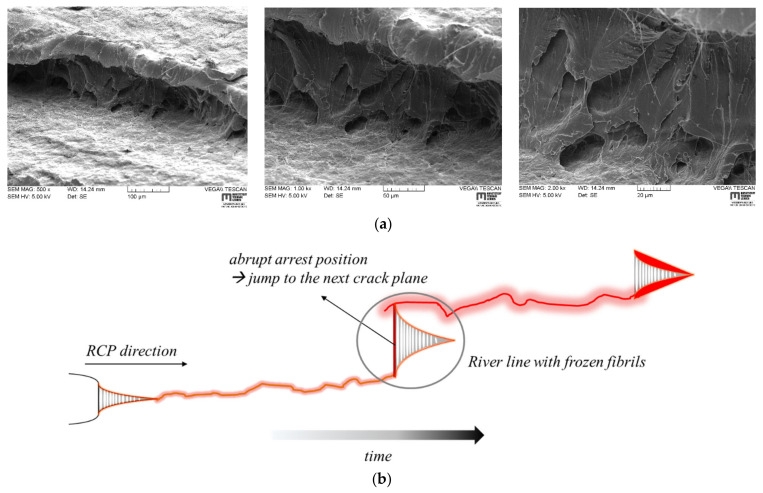
Fibrillar structures beneath river lines with increasing magnification (**a**) and schematic representation of how these structures could have evolved (**b**).

**Figure 9 materials-14-05899-f009:**
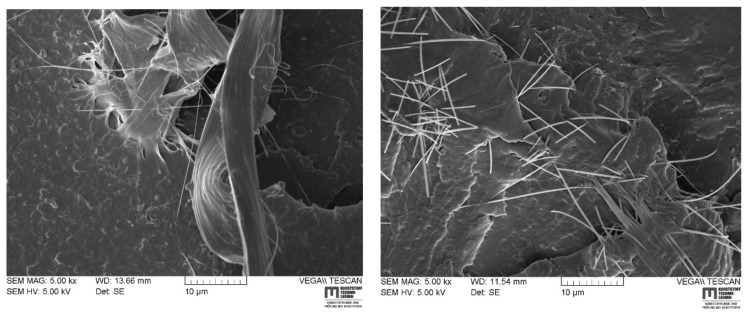
Characteristic macro—(left) and micro-fibrillar (right) structures found in all PA12 grades along the crack path after rapid fracture.

**Table 1 materials-14-05899-t001:** Basic characterization of selected PA12 grades.

Material	IM	PGM	*M* _W,rel_
**PA12-0**	−	−	1.0
**C2-im-pgm**	+	+	1.2
**C3-nc**	−	−	1.4
**C4-im-nc**	+	−	1.5
**C5-im-pgm**	+	+	1.6

**Table 2 materials-14-05899-t002:** Standard evaluation according ISO 13477 vs. modified evaluation of S4 test data.

	ISO 13477			Modified S4 Evaluation		
Material	*p*_c,0°C_ (bar)	*T*_c,5bar_ (°C)	*T*_c,12bar_ (°C)	*p*_c,0°C_ (bar)	*T*_c,5bar_ (°C)	*T*_c,12bar_ (°C)
**PA12-0**	3.16	22.5	*	3.24	22	^*^
**C2-im-pgm**	3.15	20.5	27.5	2.84	19	26
**C3-nc**	8.01	*	22–29.5	8.21	−44.5	23
**C4-im-nc**	4.08	22.5	23	4.22	15.5	21.5
**C5-im-pgm**	4.41	*	*	4.45	4.5	21

* No RCP/crack arrest transition detectable at given pressure or temperature levels.

**Table 3 materials-14-05899-t003:** Fracture surface analysis of S4 tested PA12 pipe fragments via SEM: macroscopic overview (a), river line regions (b) and brittle rapid fracture regions (c).

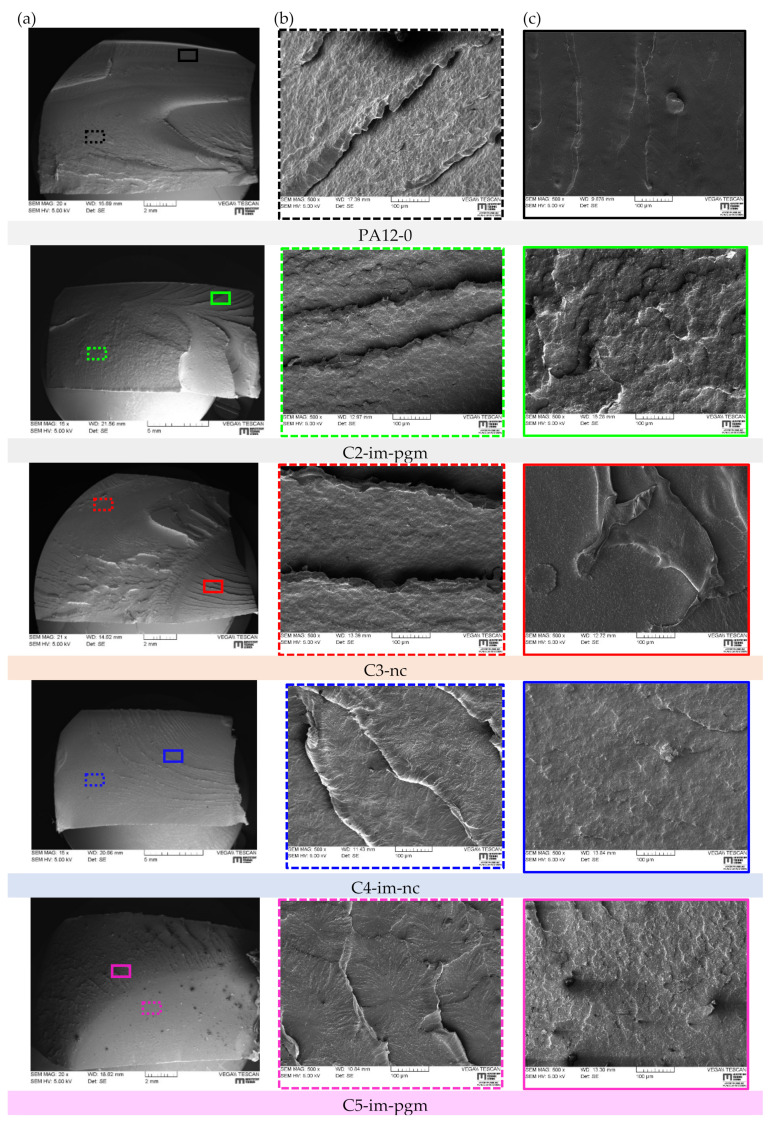

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
