# Peer review of "Structure-Property Relationships of Polyamide 12 Grades Exposed to Rapid Crack Extension"

_materials, 2021, doi:10.3390/ma14195899_

Round 1
Reviewer 1 Report
Paper presented the investigation of crack propagation in different polyamide grades. It's well written and organized, and research was fully supported by experimentation, analysis and microscopical observations. It shows a good methodology and approach for an experimental investigation. For that reason the paper is reccomended for publication
Author Response
Dear Reviewer,
thank you very much for your support!
Kind regards,
Mario Messiha
Reviewer 2 Report
Regarding the manuscript entitled "Structure-property relationships of polyamide 12 grades exposed to rapid crack extension" the subject is interesting and innovative. The manuscript is generally well written, the results are well reported, and the authors presented an accurate discussion supported by the literature. Good applications. I consider this manuscript acceptable for publishing, nevertheless throughout the text, some minor mistakes are found; then, a language revision is suggested.
Author Response
Dear Reviewer,
thank you for your time and interest, as well as for comments and recommendations! As requested we have re-read the paper again and checked for some minor language errors.
Kind regards,
Mario Messiha
Reviewer 3 Report
I have read the paper carefully and I consider a good work suitable for publication. Rapid Crack Propagation (RCP) phenomenon is a critical aspect for pipes performance that must be seriously taken into account.
ISO 13477 Small-Scale Steady-State (S4) test is the proper method for evaluating the RCP phenomenon and correctly chosen to analyze the RCP process in this work. I would like to point out the interest the novel evaluation concept of the S4 results to reduce the amount of test for determining the critical temperature and pressure value. Nevertheless, it would be necessary to analyze a greater number of samples in order to confirm the reliability of this new proposed methodology.
I understand that it may be due to confidentiality reasons but I think it would be appropriate for Table 1 to show absolute molecular weight values instead of the relative ones depending on the base material used (PA12-0).
While the fracture mechanisms involved in Slow Crack Growth (SCG) processes in pipes have been deeply analyzed and reported, in the case of the RCP process they are not yet entirely clear or unified by the scientific community. In this sense, this work also provides light through fractographic analysis reported.
In conclusion I consider this work a good paper and totally recommendable to be published in your scientific journal.
I consider correct the use of English, just to point out some errors:
Line 231 – Have to rotate instead of has to rotate
Line 311 – fractographic instead of ffractographic
Author Response
Dear Reviewer,
thank you for your time and interest, as well as for comments and recommendations!
Regarding the S4 test to validate the results with a higher reliability I can only say that we have already tested another set of 5 materials which showed similar results - more information was gained by applying our modified evaluation. However, due to confidentiality reasons I am not able to publish all of our results and as you have correctly noticed this is also the reason why we had to hide the for example the absolut values of MW for the selected pipe grades.
As requested we have re-read the paper again and checked for some minor language errors.
Kind regards,
Mario Messiha
